# Association between Patients’ Body Mass Index and the Effect of Monophasic Pulsed Microcurrent Stimulation on Pressure Injury Healing

**DOI:** 10.3390/biomedicines11092379

**Published:** 2023-08-25

**Authors:** Yoshiyuki Yoshikawa, Noriaki Maeshige, Atomu Yamaguchi, Mikiko Uemura, Terutaka Hiramatsu, Yoriko Tsuji, Hiroto Terashi

**Affiliations:** 1Department of Rehabilitation, Faculty of Health Sciences, Naragakuen University, Nara 631-0003, Japan; 2Department of Rehabilitation Science, Kobe University Graduate School of Health Sciences, 7-10-2 Tomogaoka, Kobe 654-0142, Japan; 3Department of Rehabilitation, Faculty of Health Sciences, Kansai University of Welfare Sciences, Kashiwara 582-0026, Japan; 4Department of Rehabilitation, Hosenka Hospital, Ibaraki 567-0067, Japan; 5Unit of Podiatric Medicine, Department of Plastic Surgery, Kobe University Graduate School of Medicine, Kobe 650-0017, Japan; 6Department of Plastic Surgery, Kobe University Graduate School of Medicine, Kobe 650-0017, Japan

**Keywords:** body mass index, monophasic pulsed microcurrent, pressure injury

## Abstract

This secondary analysis study aimed to detect individual variables that influence the efficacy of monophasic pulsed microcurrent on pressure injury healing. Eleven patients with pressure injuries showing delayed healing underwent a microcurrent stimulation period and a placebo period. We analyzed the correlation between the individual variables and the following three outcomes using monophasic pulsed microcurrent: the wound reduction rate in the electrical stimulation period, the reduction rate in the placebo period, and the difference between these two reduction rates. Furthermore, the patients were divided into two groups, one with a wound reduction rate of more than 10% and the other with less than 10%, and the relationship between each variable was compared. As a result, the wound reduction rate in the electrical stimulation period and the difference in the reduction rate between the two periods showed significant positive correlations with patients’ body mass index. In addition, a significant difference was observed in the body mass index between subjects with a reduction rate of 10% or higher and those with a reduction rate of less than 10%. This study found a correlation between the effect of monophasic pulsed microcurrent for pressure injury healing and the level of patients’ body mass index.

## 1. Introduction

Wound healing is divided into a series of phases, including hemostasis, inflammation, proliferation, wound shrinkage, and remodeling [1]. Skin continuity and function are restored through normal healing. Normal repair of the adult skin results in the formation of permanent scars; however, abnormal healing processes result in excessive healing, increased connective tissue deposition, the formation of hypertrophic or keloid scars, or inadequate healing. Pressure injuries may result in incomplete new tissue formation due to insufficient connective tissue deposition, leading to the formation of chronic wounds [2].

Pressure injuries are chronic wounds developed by prolonged and/or repeated external force on the local skin and underlying soft tissue with bony prominences [3]. These wounds are prevalent in patients with limited mobility, including the elderly and individuals with spinal cord injuries [3]. The severity of pressure injuries is determined through the depth of tissue damage, and the healing period is further extended due to the development of undermining ulcers with stage III or higher [3]. Such prolonged healing processes can result in increased hospitalization periods and care expenses, thereby placing a substantial economic burden on the healthcare systems [4]. Therefore, expediting the healing of pressure injuries is essential.

Electrical stimulation therapy has garnered recognition in pressure injury treatment guidelines as a modality that promotes wound healing [3]. Systematic reviews and meta-analyses [5] have substantiated its effectiveness for wound healing, considering it one of the most promising treatments. Although a number of stimulation conditions exert adverse effects, such as erythema, monophasic pulsed microcurrent (MPMC) has not been associated with these effects [6]. As a mechanism of its wound-healing effect, MPMC has been shown to promote fibroblast migration [7] and proliferation [8]. Interestingly, our clinical trial has demonstrated a significant reduction in wound size using MPMC in pressure injury patients with undermining [9]. Undermining is defined as the region underneath the overlying loose skin around a pressure injury. [10]. Meanwhile, the effect of MPMC may be influenced by patients’ individual factors affecting the condition of the tissue and cells in the wounds; scrutinizing physical factors that impact the clinical efficacy of MPMC is therefore essential for optimal clinical treatment. Hence, the present study performed an analysis using our previous clinical trial data [9], focusing on the relationship between wound healing and patients’ physical factors to explore the individual factors that affect the effectiveness of MPMC.

## 2. Methods

### 2.1. Subjects and Wounds

The inclusion criteria for this study were patients with pressure injuries with an undermining (DESIGN-R^®^ tool [11] score ≥ 15 points, National Pressure Ulcer Advisory Panel [NPUAP] stage >3) who received standard care for >2 months but whose wounds had not healed. In contrast, we excluded (1) patients without an undermining; (2) patients with malignant tumors; (3) patients with significant infection at the decubitus site; (4) patients with arterial and venous thrombosis and thrombophlebitis; (5) patients whose fever was not caused by the pressure injury and whose general condition was judged as unstable; (6) patients with anxiety about electrical stimulation; (7) patients with osteomyelitis and pressure injury necrosis; and (8) patients with other medical conditions based on which the physician deemed them unsuitable for electrical stimulation therapy (e.g., individuals with cardiac pacemakers or other bioelectrical stimulators). Information of each recruited participant has been presented in Appendix A. This study was approved by the Naragakuen University Ethics Committee (4-H028). Informed consent was obtained from all individual participants included in this study.

### 2.2. Standard Care

For periodical pressure relieving and dispersal, postural change, positioning, and mattress selection were performed in accordance with the advisory panel’s guidelines [3,9]. Pressure injury wounds were cleansed once a day with a weakly acidic detergent and the undermining area was washed using an injector with tap water. The ointments and coverings were applied according to the amount of exudate. No infections occurred throughout the study period.

### 2.3. Nutrition Management

Nutritional assessment of each participant was performed by physicians and dietitians using the Mini Nutritional Assessment-Short Form (MNA-SF), body weight, body mass index, and serum albumin levels. The MNA-SF has been validated as a nutritional assessment tool for older adults [12]. Based on these assessments, caloric requirements were calculated using the Harris–Benedict formula. The calorie intake of the participants was 1600 kcal/day for 4 individuals on oral nutrition, 1230 kcal/day for 5 individuals on central venous nutrition, and 1000 to 1200 kcal/day for the remaining 2 individuals on tube feeding. There were no changes in nutritional management throughout the study period for any of the participants.

### 2.4. MPMC Stimulation

MPMC stimulation was performed with an electrical stimulator (Ito iPES; Ito Co., Ltd., Kawaguchi, Japan). Low-ionization tendency and gold-plated rod-shaped electrodes were used for the different electrodes. The size of the electrodes was 20 mm in length and 1 mm in diameter. Five cm × five cm attachable electrodes were used for the indifferent electrodes. To perform MPMC stimulation, the different electrode was covered with sterilized gauze saturated with physiological saline solution, and if the undermining was extensive enough to allow electrode insertion, the electrode was inserted into the undermining. Once the undermining was resolved, the electrode was placed on the wound surface. The indifferent electrode was positioned on the healthy skin towards the undermining. MPMC stimulation (frequency 2 Hz, pulse width 250 ms, stimulation intensity 170 µA, and duty factor 50%) was performed once a day for 60 min, six times a week. Placebo stimulation was administered identically to MPMC stimulation, but with a 0 µA intensity.

### 2.5. Wound Evaluation

Wound evaluation was conducted twice a week using DESIGN-R and wound area (WA) measurement. DESIGN-R assessed the seven following elements: depth, exudate, size, inflammation/infection, granulation tissue, necrotic tissue, and undermining. To calculate the WA, the wound area was ascertained by adding the wound and undermining areas utilizing the tracing film (Visitrack grid^®^; Smith & Nephew Co., Ltd., London, UK) and a wound area measuring device (Visitrack^®^; Smith & Nephew Co.). The wound reduction rate was calculated as: 100 × (WA before − WA after)/WA before.

### 2.6. Blinding

All subjects, interveners, and data analysts underwent blinding. The individuals who set up the equipment for each group were the only individuals aware of the group allocation, and they were not involved in the intervention nor the data analysis. The intensity of the MPMC therapy administered in this study was less than 200 µA, which is below the sensitivity threshold, enabling the blinding of the subjects.

### 2.7. Analysis

Statistical analysis was performed using Easy R (EZR; Saitama Medical Centre, Jichi Medical University, Saitama, Japan) [13]. All data were checked for normal distribution using the Shapiro–Wilk test. The reduction rates of the wound area, including the undermining in the placebo and electrical stimulation periods, were compared using Student’s *t*-tests. Pearson’s product-moment correlation coefficient was employed to investigate the relationship between BMI, initial WA, calorie intake, serum albumin level, serum hemoglobin, C-reactive protein (CRP), DESIGN-R, and duration of the illness for three data sets: the WA reduction rate in the electrical stimulation period (E-reduction rate), the WA reduction rate in the placebo period (P-reduction rate), and the difference between the E-reduction rate and the P-reduction rate (E-P reduction rate). Furthermore, the Student’s *t*-test was used to compare the differences in age, BMI, initial area, calorie intake, serum albumin level, serum hemoglobin, CRP, DESIGN-R, and duration of the illness between the two groups based on the difference in the reduction effect of the wound size in the E-P reduction rate, which was categorized as a difference of 10% or more and less than 10%. The differences by sex and wound site (sacrum or other sites) were compared using the Chi-square test.

## 3. Results

As previously reported, the P-reduction rate was 8.2 ± 11.4% and the E-reduction rate was 26.0 ± 19.0%, with the E-reduction rate exhibiting a significant level of superiority (*p* < 0.01) (Table 1) [9]. This secondary analysis showed no significant correlation between the P-reduction rate and the background factors (BMI; r = 0.01, wound area at the start; r = −0.12, calorie intake; r = 0.17, serum albumin level; r = 0.27, serum hemoglobin; r = 0.21, CRP; r = −0.21, DESIGN-R; r = 0.05, and history of disease; r = −0.49). On the other hand, the E-reduction rate was found to only be positively correlated with the BMI (r = 0.74, *p* < 0.01), while the other background factors were not significantly correlated (wound area at the start; r = −0.34, calorie intake; r = 0.59, serum albumin level; r = −0.06, serum hemoglobin; r = −0.18, CRP; r = −0.03, DESIGN-R; r = 0.04, and history of disease; r = −0.56) (Table 2). The E-P reduction rate also showed a positive correlation with the BMI (r = 0.76, *p* < 0.01), but did not correlate with the other factors (wound area at the start; r = −0.29, calorie intake; r = 0.10, serum albumin level; r = −0.23, serum hemoglobin; r = −0.31, CRP; r = −0.10, DESIGN-R; r = 0.01, and history of disease; r = −0.28) (Table 2). Furthermore, comparisons separating subjects with a > 10% E-P reduction rate and those with ≤10% only showed significant differences with the BMI (<10% vs. 10%≤ = 18.9 ± 0.4 vs. 16.8 ± 1.0, respectively; *p* < 0.01) (Table 3). Figures of the wounds before and after MPMC treatment are provided in Appendix A.

## 4. Discussion

In this investigation, we aimed to find the relationship between the individual factors and the efficacy of MPMC therapy on pressure injuries with undermining. Our previous study [9] demonstrated that MPMC therapy was effective for patients with pressure injuries that have undermining. However, the effect exerted via MPMC varied among individuals, and some patients experienced more notable improvements than others. Therefore, we retrospectively analyzed the data in the previous study to identify the background factors that contribute to this variability. Our findings revealed that there was no significant correlation between the P reduction rate and any background factors. However, a positive correlation was observed between the BMI and the E reduction rate, or E-P reduction rate. In addition, in the investigation of background factors for the two groups of subjects (E-P reduction rates ≥ 10% or <10%), a significant difference was only observed in the BMI. These results suggest that MPMC is more effective in patients with a higher BMI. Notably, there were no significant correlations between the wound reduction rate and the factors known to affect wound healing, such as the calorie intake [14], serum albumin value [15], or serum hemoglobin value [16]. These results suggest that the BMI may be an independent factor that affects the efficacy of MPMC therapy. All subjects in this study had a BMI of 20 kg/m^2^ or less, indicating that none of the subjects appeared to be obese. The BMI is a measure of body mass that considers the weight and height of an individual, and a high BMI typically indicates obesity. However, since the subjects spent most of their time in bed under malnutrition, there were no obese patients. In addition, all participants had decreased motor function and were expected to have disuse muscle atrophy due to bed rest for more than 2 months [17]; therefore, the differences in BMI were unlikely to be due to differences in muscle mass, suggesting a potential relationship between the effect of MPMC and the existence of adipose tissue. Due to the small number of subjects in this study, we were unable to examine the relationship between the BMI and MPMC and nutritional conditions, such as calorie intake, serum albumin, and serum hemoglobin levels, but as previous studies have reported that a good nutritional status has a positive effect on wound healing [18], the interaction between MPMC and nutritional conditions may contribute towards the promotion of wound healing. In addition, ketogenic diets are known as a nutritional therapy, in which energy is obtained from lipids, while carbohydrate intake is controlled [19]. An animal study showed that ketogenic diets promoted wound healing through regulating collagen deposition and promoting proliferation and angiogenesis [20]. So far, MPMC has been believed to have positive effects on wound healing by promoting fibroblast migration towards the cathode [7] and cell differentiation [8] into myofibroblasts. An investigation of the association between the BMI and MPMC may offer novel insights into its mechanism.

Zhang et al. [21] have demonstrated the significance of dermal adipose tissue in hair growth, wound healing, and thermoregulation. In addition, dedifferentiation of dermal adipocytes into myofibroblasts might contribute to wound healing [22]. On the other hand, low-frequency electrical stimulation enhances fat metabolism [23]. Furthermore, adipose tissue has been reported to play a critical role in regulating inflammation and repair post-skin damage [24]. Research related to the effect of electrical stimulation on adipocytes, wound healing, and fat metabolism has been separately conducted in previous studies. Badhe et al. suggested that the low-intensity current stimulation of subcutaneous adipose-derived stem cells in subcutaneous adipose tissue may accelerate wound healing [25]. While the involvement of fat metabolism regulation in the effect of MPMC stimulation is unclear in the present study, it could contribute towards the effective use of MPMC for pressure injury healing. In the future, elucidating the link between MPMC and adipocytes can aid in identifying the appropriate electrode attachment sites and target tissue for MPMC therapy for wound healing. This study is a brief report using the data obtained from a previous investigation, resulting in a small sample size where more consideration of cutoff values for BMI is needed. Moreover, fat and muscle mass were not measured. The present study was conducted retrospectively and further studies using the waist-hip ratio are required in the future to address these limitations.

In conclusion, this study investigated the effectiveness of MPMC in reducing the wound area in patients with pressure injuries with pockets and the physical factors affecting the clinical efficacy of MPMC, confirming a correlation between the wound area reduction rate and the MPMC and BMI. These results suggest that MPMC may be effective in wound healing via adipose tissue. Future clarification of these mechanisms may help in selecting more appropriate pressure injury treatment for each patient.

## Figures and Tables

**Table 1 biomedicines-11-02379-t001:** Difference in the reduction rate between the placebo period and the electric stimulation period.

	Placebo Period	Electric Stimulation Period	*p*-Value
Reduction rate (%)	8.2 ± 11.4	26.9 ± 19.0	0.009

Notes: Reduction rate: 100 × (wound area at the beginning of the treatment period − wound area at the end of the treatment period)/wound area at the beginning of the treatment period.

**Table 2 biomedicines-11-02379-t002:** Correlation between the reduction rate in each period and the physical variables.

	BMI	Wound Area at Start	Calorie Intake	Serum Albumin Level	Serum Hemoglobin	CRP	DESIGN-R	History of Disease
E-reduction rate	0.74 **	−0.34	0.59	−0.06	−0.18	−0.03	0.04	−0.56
P-reduction rate	0.01	−0.12	0.17	0.27	0.20	−0.21	0.05	−0.49
E-P reduction rate	0.76 **	−0.29	0.10	−0.23	−0.31	0.10	0.01	−0.28

Abbreviations: E-reduction rate: wound area reduction in the electrical stimulation period, P-reduction rate: wound area reduction in the placebo period, E-P reduction rate: difference between the E-reduction rate and the P-reduction rate, BMI: body mass index, and CRP: C-reactive protein. ** *p* < 0.01.

**Table 3 biomedicines-11-02379-t003:** Comparison of differences in each item in two groups divided according to the wound reduction rate.

	<10%	10%≤	*p*-Value
BMI (kg/m^2^)	16.8 ± 1.0	18.9 ± 0.4	0.008
Wound area before the study (cm^2^)	14.2 ± 13.1	9.1 ± 4.0	0.82
Calorie intake (kcal)	1198 ± 242.9	1338 ± 1.203.0	0.74
Serum albumin level (g/dL)	2.9 ± 0.3	2.8 ± 0.6	0.63
Serum haemoglobin (g/dL)	10.7 ± 2.1	10.6 ± 2.0	0.69
CRP (mg/dL)	2.5 ± 3.1	2.0 ± 2.2	0.69
DESIGN-R (Score)	20 ± 4.2	21 ± 3.2	0.57
History of disease (Months)	9.0 ± 7.0	7.3 ± 4.4	0.63
Sex (male/female)	2/3	2/4	0.81
Pressure injury (sacral/non-sacral)	3/2	3/3	0.74

Abbreviations: BMI: body mass index, and CRP: C-reactive protein.

## Data Availability

The data that support the findings of this study are available on request from the corresponding author. The data are not publicly available due to [restrictions, e.g., their containing information that could compromise the privacy of research participants].

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
