# Peer review of "Association between Patients’ Body Mass Index and the Effect of Monophasic Pulsed Microcurrent Stimulation on Pressure Injury Healing"

_biomedicines, 2023, doi:10.3390/biomedicines11092379_

Round 1
Reviewer 1 Report
The authors use data from a previous study to demonstrate that BMI correlates with efficacy of MPMC for the treatment of pressure injuries. While the concept is interesting, the presentation of the results is so brief that it is difficult to grasp the significance of the results. The authors just say there is a correlation with a results section that is 7 lines. I feel that the results can be presented in a much more detailed fashion.
The correlation with BMI seems to be an indicator of malnutrition that impairs healing in the low BMI group. There is a little bit of discussion about this concept, but it could be expanded. In other words, despite their indices of malnutrition, a low protein mass inhibits the activities of the MPMC. Adequate nutritional status is needed for effect.
I would also expect that the E-P reduction rate would have to correlate with BMI, since only the larger BMI had an effect (the two are linked).
Reviewer 2 Report
The paper entitled “Association between Patients’ Body Mass Index and the Effect of Monophasic Pulsed Microcurrent Stimulation on Pressure Injury Healing” includes potentially relevant data for design and development innovatory regenerating strategies directed toward tissue damage with impaired, prolonged repair. The Authors of this publication indicated among others that “…significant difference was observed in body mass index between subjects with a reduction rate of 10% or higher and those with a reduction rate of less than 10%...”.
Remarks:
1. it is difficult to find in the “Introduction” subchapter information (even brief) on the subsequent phases of the healing process, i.e. hemostasis, inflammation, proliferation and remodeling;
2. the article lacks information about the inclusion and exclusion criteria of the study - this absolutely should be completed;
3. the article lacks information about the bioethics committee's approval of the experiment - this needs to be added.
4. information on nutrition is laconic and the standardization of the procedure described in subsection “2.3 Nutrition Management” has not been cited.
5. statistical methods are described properly, but no justification is given for the choices made - which may be a reason for concern about reliable statistical control of the results obtained; moreover, relevant literature references are not indicated.
6. an important part of the discussion section (lines 175-183) was deprived of comparison/reference to the results of other authors' studies - should be completed
7. the conclusion is laconic and does not fully answer the (scientific) question stated in the purpose of the work - should be modified.
8. an anthropometric index (such as BMI) has numerous drawbacks and limitations (e.g. age, sex, ethnicity, and muscle mass can influence the relationship between BMI and body fat; BMI does not distinguish between excess fat, muscle, or bone mass, nor does it provide any indication of the distribution of fat among individuals) - what is the reason for omitting the WHR?
9. what is the reason for the lack of evaluation of insulin resistance or insulin sensitivity and their impact on impaired wound healing?
Reviewer 3 Report
The article under review is devoted to the clinical observation of the effectiveness of monophasic pulsed microcurrent stimulation in the treatment of pressure injuries. The topic is relevant.
The article is essentially written based on the results of an already published study [7]. Just changed the names of the parameters, so in the published article S-period presented in Biomedicines - P-rate reduction. The proposed methods are one repetition of past events, since the work is the same. Accordingly, the data are the same as in [7] processed by other methods (Tables 1, 2).
With this observation period, in the [7] - 12, in present article -11. Why?
If the authors tried a meta-analysis of other authors' data on this topic, it would be listening to the work amplifier, but, unfortunately, in this form, this is a short message, but by no means a complete article.
Reviewer 4 Report
Dear Authors,
The manuscript entitled "Association between Patients’ Body Mass Index and the Effect of Monophasic Pulsed Microcurrent Stimulation on Pressure Injury Healing" represents a valuable study in the field.
Only minor revisions i have for the submitted manuscript.
1) The authors should provide patients individual characteristics such as peripheral blood examination (WBCs, RBCs,HCT, etc) and biochemical examination as supplementary material.
2) Could the authors provide any figures of the wounds?
The language is fine only minor corrections are required.
Reviewer 5 Report
Very high quality article. Excellent findings ! You can improve it by discussinng something not oftenly adressed the impact of diet on PAD. Please check: Implicating the effect of ketogenic diet as a preventive measure to obesity and diabetes mellitus. Life Sci. 2021 Jan 1;264:118661. doi: 10.1016/j.lfs.2020.118661. Epub 2020 Oct 26. PMID: 33121986.
Language is ok.
Reviewer 6 Report
In this study, a correlation was found by the authors between the effect of monophasic pulsed microcurrent for pressure injury wound healing and body mass index. Please see the attached pdf for comments.

Round 2
Reviewer 1 Report
Accept
Author Response
Thank you for your valuable evaluation of our paper.
Reviewer 2 Report
The Authors of the manuscript entitled: "Association between Patients' Body Mass Index and the Effect of Monophasic Pulsed Microcurrent Stimulation on Pressure Injury Healing", submitted to editorial system of Biomedicines modified appropriately the text of the article.
Article meets all the criteria to be published without any corrections.
Author Response

(The authors gave the same response as above.)

Reviewer 3 Report
The authors made changes to the article, significantly improved the materials and methods section. The meaning of the obtaine a new ethical approval is not entirely clear, since no new studies have been conducted, the data of the past study were taken. Changing the composition of the research group also does not change the data obtained.
Undoubtedly, the result obtained by the authors is interesting. However, since all the data have already been published, this result is not sufficient to write a full-fledged article. If there were confirmation, for example in vitro, of the positive effect of electrical stimulation of adipocytes on regeneration, this would significantly complement the study. Or, as I have already noted, to analyze the already published results of the influence of body mass index on the healing of bedsores. There are many such works in the literature (https://journals.lww.com/jncqjournal/fulltext/2009/04000/Body_Mass_Index,_Weight,_and_Pressure_Ulcer.8.aspx; https://www.liebertpub.com/doi/abs/10.1089/ wound.2020.1275). I agree that the work uses the technique of low-frequency stimulation of regeneration, and this is a novelty of the work, but the evidence is not enough for the conclusions drawn. Undoubtedly, fat cells can differentiate into myofibroblasts, but the issue with low BMI is more likely to affect the overall compensatory capabilities of the body, with low BMI - significant depletion.
Therefore, I believe that the result is interesting, but there is not enough data for a full article, since the data completely repeats the already published article.
It is not entirely clear why the piece about the keto diet is inserted. These are controversial issues, so there are recommendations with the mandatory presence of carbohydrates for successful wound healing (https://www.magonlinelibrary.com/doi/abs/10.12968/bjon.2001.10.Sup1.5336)
Author Response
We appreciate the time and effort you have dedicated to providing insightful feedbacks. We showed the comments from you and our response below.
Comments
・The meaning of the obtaine a new ethical approval is not entirely clear, since no new studies have been conducted, the data of the past study were taken. Changing the composition of the research group also does not change the data obtained.
・Since all the data have already been published, this result is not sufficient to write a full-fledged article. If there were confirmation, for example in vitro, of the positive effect of electrical stimulation of adipocytes on regeneration, this would significantly complement the study. Or, as I have already noted, to analyze the already published results of the influence of body mass index on the healing of bedsores. There are many such works in the literature (https://journals.lww.com/jncqjournal/fulltext/2009/04000/Body_Mass_Index,_Weight,_and_Pressure_Ulcer.8.aspx; https://www.liebertpub.com/doi/abs/10.1089/ wound.2020.1275).
・Therefore, I believe that the result is interesting, but there is not enough data for a full article, since the data completely repeats the already published article.
(Response)
Thank you for the comments. As you advised, a secondary analysis with the data from multiple reports is more relevant for the publication. Meanwhile, despite plenty of papers on the relationship between fat mass and wound healing, there are no papers other than our previous study investigating the relationship between the effect of ES on wound healing and the patient's BMI. Therefore, the present study includes only the data from our previous paper. Although we added a description of our previous data in the Result section according to the suggestions from other reviewers, we listed it as a citation in order to distinguish the new analysis data from previous data. In addition, we have changed the type of paper from "Article" to "Brief report" according to the advice from the section managing editor. We believe this brief report of a secondary analysis could contribute to future research.
Comments
・If there were confirmation, for example in vitro, of the positive effect of electrical stimulation of adipocytes on regeneration, this would significantly complement the study.
(Response)
We agree with your comment. I already have started the project using cultured cells; however, we have not obtained the data yet. We would like to submit a paper on this topic as a different paper in the future.
Comments
・I agree that the work uses the technique of low-frequency stimulation of regeneration, and this is a novelty of the work, but the evidence is not enough for the conclusions drawn.
(Response)
Thank you for the comments. I understand the limitation of our paper according to the comments from reviewers, and we avoided over-expression in the conclusion in the revised manuscript we submitted last time. I would appreciate it if you could consider this point.
Comments
・It is not entirely clear why the piece about the keto diet is inserted. These are controversial issues, so there are recommendations with the mandatory presence of carbohydrates for successful wound healing
(Response)
Thank you for the comments. This revision is according to the suggestion from another reviewer and the revised description was approved. To avoid over-expression on this topic, I described this information as a report from an animal study in the revised manuscript. (Line 206-207)
Round 3
Reviewer 3 Report
I respect the authors and the work done, but my opinion is that it is not correct to publish an article based on the same results.